# Effect of Biosynthesized Silver Nanoparticles on the Growth of the Green Microalga *Haematococcus pluvialis* and Astaxanthin Synthesis

**DOI:** 10.3390/nano13101618

**Published:** 2023-05-11

**Authors:** Petras Venckus, Ieva Endriukaitytė, Kotryna Čekuolytė, Renata Gudiukaitė, Andrius Pakalniškis, Eglė Lastauskienė

**Affiliations:** 1Institute of Biosciences, Life Sciences Center, Vilnius University, Sauletekio Ave. 7, LT-10257 Vilnius, Lithuania; ieva.endriukaityte@gmc.stud.vu.lt (I.E.); kotryna.cekuolyte@gmc.vu.lt (K.Č.); renata.gudiukaite@gf.vu.lt (R.G.); 2Institute of Chemistry, Vilnius University, Naugarduko 24, LT-03225 Vilnius, Lithuania; andrius.pakalniskis@chgf.vu.lt

**Keywords:** silver nanoparticles, *Haematococcus pluvialis*, astaxanthin

## Abstract

Silver nanoparticles (AgNPs) are widely known for their antimicrobial activity in various systems from microorganisms to cell cultures. However, the data on their effects on microalgae are very limited. Unicellular green algae *Haematococcus pluvialis* is known for its ability to accumulate large amounts of astaxanthin under stress conditions. Therefore, it can be used as a suitable model system to test the influence of AgNPs on stress induction in unicellular algae, with visible phenotypic effects, such as astaxanthin synthesis and cell morphology. This study tested different AgNP concentrations (0–8 mg/L) effects on different growth stages (red and green) of *H. pluvialis* culture. Effects on cell morphology, culture productivity, and astaxanthin synthesis were evaluated. Data showed that the addition of high concentrations of AgNPs to the growing culture had a significant negative impact on culture productivity. Green-stage (HpG) cultures productivity was reduced by up to 85% by increasing AgNPs concentration to 8 mg/L while the impact on red-stage (HpR) culture was lower. Astaxanthin concentration measurements showed that AgNPs do not have any effect on astaxanthin concentration in HpG culture and caused decreased astaxanthin production rate in HpR culture. HpG culture astaxanthin concentration stayed constant at ~0.43% dry weight, while HpR culture astaxanthin concentration was significantly reduced from 1.89% to 0.60% dry weight by increasing AgNP concentration. AgNPs in the media lead to significant changes in cell morphology in both HpG and HpR cultures. Cell deformations and disrupted cytokinesis, as well as AgNPs and induced sexual reproduction, were observed.

## 1. Introduction

Metal (Ag) or metal oxide nanoparticles, such as iron oxide (Fe_3_O_4_), titanium oxide (TiO_2_), copper oxide (CuO), zinc oxide (ZnO), etc., are known to have antimicrobial activity [1,2]. Silver nanoparticles (AgNPs) have been used in medicine as bandages and other medical tool components to keep them sterile and prevent the infection and inflammation of wounds [3]. AgNPs have a stronger effect and can release silver ions faster compared to silver metal coating due to their larger surface and smaller particle size [4]. AgNPs have a complex effect on cells. The toxicity levels of AgNPs, as well as other metal nanoparticles (NPs), could be different depending on their shape, size, coating, synthesis method, charging, internal properties, and the type of cell that is exposed [5,6,7,8,9]. Cellular uptake mechanisms of nanoparticles are similar in bacteria and unicellular algae [10]. Positive metal ions first bind to cell wall functional groups such as carboxyl, phosphate, hydroxyl, amino, etc. Then, they are absorbed through the cell wall and localized in various organelles [10]. The antibacterial action of AgNPs occurs when Ag^+^ ions are released and bind to the bacteria cell proteins and DNA, thus disrupting their functionality and generating reactive oxygen species (ROS) [11].

AgNPs can be obtained by chemical, physical, or biological synthesis. Biological synthesis has gained more attention due to being an environmentally friendly and inexpensive method. Compared to chemical and physical synthesis methods, biological methods do not require high temperatures or expensive equipment, do not leave toxic waste, etc. [12]. Both mesophilic and thermophilic *Bacillus* sp. and *Geobacillus* sp. bacteria are used for the biological synthesis of AgNPs and other metal nanoparticles. It has been shown that supernatants of mesophilic *Bacillus subtilis* with added 1 mM of AgNO_3_ can synthesize 3–20 nm diameter AgNPs [13]. Deljou and Godarzi (2016) used thermophilic *Bacillus* sp. supernatants with added AgNO_3_ to obtain 7–31 nm AgNPs [14]. Both obtained AgNPs showed antimicrobial activities against Gram-positive and Gram-negative pathogenic bacteria [12,13,14]. *Geobacillus stearothermophilus* secretome with added 1 mM concentration of AgNO_3_ resulted in the extracellular biological synthesis of 5–35 nm diameter AgNPs [15]. Ghasemi et al. (2018) also used *G. stearothermophilus* secretomes to obtain AgNPs [13]. The mechanism of the formation of AgNPs using *Geobacillus* sp. bacteria is not clear, although it is hypothesized that secreted enzymes are important for the reduction of silver ions and the formation of AgNPs [15]. The NADH-dependent nitrate reductase enzyme is involved in the green synthesis of AgNPs, converting nitrates to nitrites and thus reducing silver ions to AgNPs [16].

Ketocarotenoid astaxanthin is widely used in medicine and aquaculture due to its anti-oxidative activity [17,18]. A naturally living organism, capable of synthesizing the largest amount of astaxanthin (5.9% of its dry biomass), is the alga *Haematococcus pluvialis* [19]. It is a unicellular, freshwater, green microalga, found in various freshwater bodies, such as fountains, artificial pools, ponds, rain pools, or birdbaths [20,21]. It has a unique cell life cycle, consisting of two different phases: motile and non-motile [22]. Motile cells are 5–30 µm in diameter, round, ovate or ellipsoid in shape, and have a thin cell wall, and two flagella, while non-motile cells are 20–50 µm in diameter, round, have a thick cell wall with a tertiary wall, and no flagellum [22,23]. The *H. pluvialis* life cycle and changes in morphology depend on environmental conditions. Under optimal conditions, *H. pluvialis* culture grows and synthesizes small amounts of astaxanthin (~0.8% dry weight), but the dominant pigments are chlorophylls, and algae appear green (green stage). Their life cycle consists of mainly small, thin-walled, motile zoospores and slightly bigger, round, non-motile aplanospores, which under favorable conditions quickly reproduce asexually by producing 2–8 zoospores [19]. Under stress-inducing conditions such as high salinity, light, or nitrogen starvation, *H. pluvialis* begins to produce and accumulate the red pigment astaxanthin as a protective agent (up to 5.7% dry weight) and algae cells appear red (red stage) [24,25]. During the red stage, the algae life cycle is longer, cells are bigger than in the green stage, mostly non-motile, due to low reproduction rate, and have thick, multi-layered cell walls hardened by cellulose [26]. In addition, using extreme conditions, such as freezing or medium chemistry changes, followed by favorable growth conditions can induce sexual reproduction when some aplanospores initiate gametogenesis and become 64-gamete-containing gametocysts [27]. Astaxanthin can be synthesized chemically; however, its properties remain inferior compared to naturally synthesized pigment [25]. Therefore, it is necessary to find ways to enhance astaxanthin biological production using *H. pluvialis*. Few studies have described AgNPs’ influence on algae and *H. pluvialis* in particular. For achieving the highest astaxanthin production using *H. pluvialis* algae, optimal stress conditions should be found. One of the proposed stressors could be AgNPs [28]. Therefore, their influence on algae growth or astaxanthin synthesis is not clear. Although the cell walls of algae are different from those of bacteria and the entry of silver ions into the cell is altered, the effect of ions is as lethal on algae cells as it is on bacteria [2]. AgNPs in *H. pluvialis* cells cause oxidative stress by ROS generation, and thus algae may start to synthesize the antioxidant astaxanthin as a protective agent [29]. Furthermore, AgNPs affect photosynthetic organisms, including algae, by disrupting chlorophyll synthesis, which could lead to cell malfunctioning, fading, and death [2].

There is little understanding of the effects of biosynthesized AgNPs on the growth of these algae, astaxanthin production, and morphology changes during different cell cycle phases. Since *H. pluvialis* and its product astaxanthin are important in the industry, more accurate research is needed. This study aimed to investigate the effect of different concentrations of AgNPs (1–8 mg/L) produced by using the *Geobacillus* sp. 25 strain on *H. pluvialis* green-stage and red-stage cultures. The effects were measured by analyzing growth, astaxanthin concentration, and cell morphology changes.

## 2. Materials and Methods

### 2.1. Organism and Culturing Conditions

The freshwater green algae *H. pluvialis* strain SAG 192.80 was obtained from the Culture Collection of Algae at Gottingen University (Germany). Two different stages of initial cultures were made. Green-stage (HpG) (0.85 ± 0.09% astaxanthin) culture was cultivated in BB medium (25 mg/L CaCl_2_·2H_2_O; 75 mg/L MgSO_4_·7H_2_O; 75 mg/L K_2_HPO_4_; 175 mg/L KH_2_PO_4_; 25 mg/L NaCl; 50 mg/L EDTA; 31 mg/L KOH; 5 mg/L FeSO_4_·7H_2_O; 1 µL H_2_SO_4_; 8.8 mg/L ZnSO_4_·7H_2_O; 1.4 mg/L MnCl_2_·4H_2_O; 0.7 mg/L MoO_3_; 1.6 mg/L CuSO_4_·5H_2_O; 0.5 mg/L Co(NO_3_)_2_·6H_2_O; and pH~6.8) [30] also containing 165 mg N/L in NaNO_3_ (to avoid N starvation conditions during the cultivation period). Red-stage (HpR) (1.7 ± 0.19% astaxanthin) was cultivated in BB medium containing 41.25 mg N/L in NaNO_3_ (substantial to ensure limited growth and high astaxanthin production, and only residual amounts of N in the media before the start of the experiment). Both cultures were produced in 250 mL Erlenmeyer flasks using appropriate BB medium under continuous white LED (16 W, 3500 K) light at 14.43 ± 0.8 µmol·m^−2^·s^−1^ and 21 °C temperature.

*Geobacillus* sp. strain 25 was isolated at the Department of Microbiology and Biotechnology, Institute of Biosciences (former Faculty of Natural Sciences), Life Sciences Center, Vilnius University (Lithuania) using samples taken from an oil well. The culture was cultivated in liquid broth with the following composition: 10 g/L tryptone (Roth, Karlsruhe, Germany) and 5 g/L meat extract (Merck, Rahway, NJ, USA); the medium was supplemented with the following salts: 5 g/L NaCl (Merck), 2.3 nM CaCl_2_ (Merck), and 0.91 µM ZnSO_4_ (Merck) in 250 mL Erlenmeyer flasks. The culture was grown in an orbital shaker at 55 °C and 180 rpm. After 48 h of incubation, cells were separated by centrifugation at 16,000× *g* for 10 min. This resulted in cell-free extracts that were used as material for AgNP synthesis.

### 2.2. Experimental Design

The dry weight and cell count of both starter cultures were determined before the experiment. For HpG culture, BB medium with 41.25 mg N/L was used, and for HpR culture, BB medium without NaNO_3_. Dilutions were made to reach an appropriate starter culture concentration of 0.2 g/L dry weight. We used extracellularly biosynthesized AgNPs. Six Erlenmeyer flasks, each containing 50 mL of culture (0.2 g/L dry weight), were supplemented with AgNPs stock to reach concentrations of 0 mg/L (control), 1 mg/L, 1.5 mg/L, 2 mg/L, 4 mg/L, and 8 mg/L (0 µL, 50 µL, 100 µL, 150 µL, 200 µL, and 400 µL of stock, respectively) in the media. Appropriate amounts of deionized water were added to ensure equal volumes of culture. Flasks were plugged with cotton plugs and cultivated for seven days. HpG samples were maintained at 21 °C temperature providing continuous white LED (16 W, 3500 K) light of 14.43 ± 0.8 µmol·m^−2^·s^−1^. HpR samples were maintained at 21 °C with continuous illumination of white LED (18 W, 3500 K) light of 112.6 ± 3.21 µmol·m^−2^·s^−1^ (a higher amount of light was set to ensure stressful conditions). Six biological replicates were cultivated during six different periods.

### 2.3. Analytical Procedures

Dry weight was determined by centrifuging a sample, drying it at 70 °C for at least two hours, and weighing it. After the dry weight measurement, the same biomass was used for astaxanthin extraction and concentration in the biomass determination. Astaxanthin was extracted, and concentration was measured using the following method: The known mass of the algae biomass (5–15 mg (depending on expected astaxanthin concentration)) was taken and mixed with 5% KOH, 30% methanol, and 70% water solution, and kept at 70 °C temperature in a water bath for 10 min to remove chlorophyll. Astaxanthin extraction was performed using 5 mL DMSO and 0.1 mL acetic acid at 70 °C in a water bath for 10 min. Extraction with DMSO was repeated till the biomass turned grey. Astaxanthin content in the extract was measured by using a spectrophotometer at 490 nm and 750 nm wavelengths [31]. The concentrations of astaxanthin (%) were calculated by using the following formula:$$A=\frac{\left(OD_{490}-OD_{750}\right)\times 5.6\times SV}{BM}\times 100\%$$
where 5.6 is the absorbance coefficient; *SV*—sample volume (L); *BM*—biomass mass (mg).

### 2.4. Microscopic Analysis

At the end of the cultivation period, samples for scanning electron microscopy (SEM) and light microscopy were taken. Samples for SEM were prepared using phosphate-buffered saline (PBS) buffer for salt removal, 3% glutaraldehyde for cell fixation, and different ethanol concentrations (50%, 70%, 80%, and two times 95%) for water removal from the cells. Micrographs were taken using a Hitachi (Japan) SEM SU-70 scanning electron microscope (SEM). The accelerating voltage was kept at 10 kV. Samples were deposited from suspension on a small piece of a silicon wafer, which was attached to the aluminum sample holder via double-sided carbon tape. To improve the samples’ conductivity, they were coated with a thin layer (5 nm) of silver by using magnetron sputtering with a Quorum (United Kingdom) Q105T ES sputter.

An Olympus BX51 microscope with Qimaging Micropublisher 3.3 RTV camera was used for taking light microscopy micrographs and for cell count analysis. Cells were counted using a Burker cell counting chamber. Different cell types were counted separately (zoospores, aplanospores, deformed or undivided cells, gametocysts, and dead cells). Three technical replicates were made for every sample. Samples were not fixed.

### 2.5. Synthesis and Characterization of Silver Nanoparticles

The *Geobacillus* sp.-25-bacteria-induced synthesis of AgNPs, the size distribution, using the dynamic light scattering method, and the Zeta potential value analysis were performed in the same way as in the Cekuolyte et al. (2023) [32]. In this study, the AgNP size distribution and the Zeta potential value were determined by keeping the obtained AgNPs in an algae culture medium for seven days (the AgNPs were rinsed with distilled water before the measurements).

### 2.6. Statistical Analysis

Analyses were conducted using R (version 4.0.5) (R Core Team (2021), Vienna, Austria). Linear regression analysis was used for the determination of the significance of the AgNP concentration effect on algae growth and astaxanthin synthesis changes. The significance level used was *p* < 0.05.

## 3. Results

### 3.1. Characterization of AgNPs in Algal Culture Medium

To determine the size and stability of AgNPs obtained using the secretomes of *Geobacillus* sp. strain 25 bacteria during the experiment lasting seven days, these AgNPs were kept for this period in the algal culture medium. The size distribution results are presented in Table 1.

The obtained results show that the size of most of the AgNPs in the algal medium is <100 nm, but compared to the results obtained by measuring the AgNP size immediately after the synthesis, there was a reduction (from 99 to 86 percent) in AgNPs with a diameter of less than 100 nm [32]. This indicated that some of the AgNPs obtained using the secretome of *Geobacillus* sp. strain 25 bacteria aggregate in the algal medium.

The Zeta potential value of the AgNPs after seven days of incubation in an algal medium was also measured. The determined Zeta potential value was −29.1 (±0.16) mV, which shifted to the positive side by 2.2 mV (from −39.3 mV) compared to the Zeta potential value of the AgNPs obtained using *Geobacillus* sp. strain 25, measured immediately after the synthesis [32]. This change in the Zeta potential indicated that the components in the algal medium are potentially binding to the surface of the AgNPs, which can lead to an increase in the number of aggregated AgNPs.

### 3.2. AgNPs’ Effect on Algae Biomass Growth and Astaxanthin Synthesis

After a seven-day cultivation period, dry weight change and astaxanthin concentration were determined for both HpG and HpR cultures. Dry weight analysis showed decreasing culture productivity with increased AgNP concentration in the media. HpG green culture’s productivity was diminished from 0.41 g/L to 0.13 g/L dry weight with increased concentration of AgNPs (0–4 mg/L) in the media (linear regression R^2^ = 0.774; *p* < 0.05). Higher concentrations (4–8 mg/L) of AgNPs greatly reduced culture growth by causing cell death and changing the cell metabolism, leading to lower biomass concentration than the initial one (0.2 g/L) (Figure 1A). On the other hand, the HpR culture, as expected, showed much lower overall productivity (Figure 1B). Accumulation of dry weight on average lowered from 0.23 g/L in control to 0.09 g/L in culture containing 4 mg/L of AgNPs. Further increase in AgNP concentration did not affect culture growth and it rarely led to cell death and lower than initial dry weight. The linear regression model (0–4 mg/L AgNPs) shows significant culture productivity dependence on AgNP concentration in the media (R^2^ = 0.241; *p* < 0.05).

Astaxanthin concentration in HpG biomass was not significantly affected by the addition of AgNPs (linear regression R^2^ = 0.123; *p* > 0.05) and remained low at 0.43% on average (initial average astaxanthin concentration in the cultures was 0.88%) (Figure 1C). Therefore, the HpR culture’s astaxanthin concentration was diminished from 2.09% (control) to 0.42% (8 mg/L AgNPs) astaxanthin in dry weight by adding AgNPs. Linear regression showed a strong dependence of astaxanthin concentration in the biomass on AgNP concentration in the media (R^2^ = 0.675; *p* < 0.05) (Figure 1D).

#### AgNPs’ Effect on Morphology and Count

Using cell count and dry weight data, average cell mass was calculated for both HpG and HpR cultures (Figure 2). We observed that increasing AgNP concentration from 0 to 2 mg/L led to decreased cell size from 1.3 × 10^−5^ g/cell to 3.6 × 10^−6^ g/cell. Further, an increase in AgNP concentration did not have an influence on cell size in the HpG culture. On the other hand, HpR culture did not show a strong response to increased AgNP concentration, and average cell size varied from 2.0 × 10^−5^ g/cell to 4.3 × 10^−5^ g/cell. Overall cell size was larger in HpR cultures.

Changes in different morphology cell counts were observed using light microscopy (Figure 3). Exposure to a different concentration of AgNPs led to morphological changes in both HpG and HpR cultures. Increasing AgNP concentration from 0 to 2 mg/L led to cell death in the HpG culture as well as an increase in the live cell count (Figure 3A). The increased cell count was due to the increased number of zoospores (Figure 4a) in the culture. The number of zoospores increased from 2.7 × 10^4^ cells/mL in the control culture to 9.9 × 10^4^ cells/mL in the culture with 2 mg/L AgNPs. Further increase in AgNP concentration led to decreased numbers of zoospores due to metal toxicity which accelerated cell death (Figure 3C). The number of aplanospores (Figure 4b,c) slowly but noticeably decreased with increasing AgNP concentration (from 1.8 × 10^4^ cells/mL 1 mg/L AgNPs to 4.5 × 10^3^ cells/mL 8 mg/L). AgNPs had a strong influence on cell shape and division. Concentrations of 2 mg/L AgNPs increased the deformed and/or undivided cell (Figure 4d) count on average fivefold, from 3.0 × 10^2^ cells/mL in control culture to 1.5 × 10^3^ cells/mL in culture with 2 mg/L AgNPs. Concentrations higher than 2 mg/L of AgNPs induced the sexual reproduction process as gametocysts (Figure 4e) were found in cultures. No gametocysts were found in cultures with no AgNPs or small amounts of AgNPs. The highest number of gametocysts (3.3 × 10^2^ gametocysts/mL) was found in an 8 mg/L AgNP culture.

In HpR cultures, all changes were much subtler and reduced. As well as in the culture, increased AgNP concentrations led to decreased numbers of living cells. No changes in the dead cell count were observed (Figure 3B). Different cell counts (Figure 3D) show a continuous decrease in zoospore (Figure 4g) and aplanospore (Figure 4h,i) cell counts with increased AgNPs concentrations (from 1.0 × 10^4^ cells/mL 0 mg/L to 2.3 × 10^3^ cells/mL 8 mg/L for zoospores and from 9.2 × 10^3^ cells/mL 0 mg/L to 3.8 × 10^3^ cells/mL 8 mg/L for aplanospores). No changes in the deformed cell count were found using different concentrations of AgNPs (from 2.2 × 10^2^ cells/mL 4 mg/L to 5.1 × 10^2^ cells/mL 2 mg/L). Gametocysts were found in small numbers (8.0 × 10^1^ gametocysts/mL) only in the culture with 8 mg/L AgNP concentration.

The results obtained during SEM analysis revealed morphological changes in both HpR and HpG cultures (Figure 5). Both culture control sample cells were mostly regular round shapes with no visible deformations (Figure 5a,e) and they remained regular after exposure to low concentrations of AgNPs (1–3 mg/L) (Figure 5b,f). HpG treatment with 2 mg/L of AgNPs caused severe cell deformations (Figure 5c). The exposure to higher concentrations of AgNPs (4–8 mg/L) in both cultures resulted in more debris, and cells were covered with film-like material (Figure 5g) or stuck together (Figure 5d,h).

## 4. Discussion

The effect of AgNPs produced by *Geobacillus* sp. cells on microalgae has not yet been analyzed. The results of growth showed a negative AgNP influence on algal cultures. Both HpG and HpR cultures showed a decrease in biomass production. Linear regression analysis for HpG culture productivity showed a strong dependence of biomass production on AgNP concentration (R^2^ = 0.679), showing that it was the primary contributor to the observed decrease in culture productivity. Productivity in these cultures stayed at 85% lower compared to control cultures. This indicates AgNPs’ negative influence on cell growth as well as on cell division. It is known that various metal NPs could have different effects on green-stage *H. pluvialis* growth. It was shown that magnesium aminoclay (MgAC) nanoparticles in concentrations up to 1 g/L have a positive effect on biomass production [33]; meanwhile, ZnO NPs have a negative impact on biomass production even in low concentrations of 50 µg/mL [34]. HpR culture showed higher resistance to the addition of AgNPs. We were able to adapt a linear regression model to describe changes in culture productivity (R^2^ = 0.241). The determination coefficient for HpR culture is lower compared to HpG, indicating higher resistance of cells to AgNPs. We observed a 50% decrease in culture productivity compared to the control culture, but it is worth noting that there was a high degree of variation between cultivation periods, indicating that starting culture conditions (mainly residual amounts of nitrogen and other elements, starting culture density, and growth phase) was the main factor influencing culture productivity. It is worth noting that even at 8 mg/L AgNP, the cultures (HpG and HpR) did not die, but also slightly increased in dry weight, and the live cell count did not drop, indicating that this concentration is not high enough to exterminate algae completely.

One of the ways nanoparticles can disrupt cell functions is by mechanically damaging the cell membrane [35]. Other green algae (*Dunaliella* and *Chlorella*) have shown similar responses to various AgNP concentrations. The concentrations of 10 mg/L of AgNPs decreased cell viability by more than 90% in both species [36]. In general, *H. pluvialis* proved to be more resistant to AgNPs compared to some non-extremophilic bacteria. These nanoparticles are effective against both Gram-positive and Gram-negative bacteria, with MIC values between 0.11 mg/L and 0.86 mg/L against *Staphylococcus aureus*, *B. subtilis*, *Pseudomonas aeruginosa*, *Acinetobacter baumannii*, and *Escherichia coli* [37]. Vijay Kumar et al. [38] obtained AgNPs using the plant extract of *Horerhavia diffusa* and tested them against three fish pathogens. The results showed that AgNPs were most effective against *Flavobacterium branchiophilum* (MIC 50 μg/mL) [38]. This can be attributed to its metabolism and cell wall composition. The *H. pluvialis* cell wall is multi-layered and consists mainly of proteins (green stage) or polysaccharides (red stage) [39]. Moreover, cell walls contain aliphatic biopolymers which are attributed to resistance to chemical compounds [40]. It was shown that the cell membrane has various proteins that can selectively transport metal ions into and out of the cell, potentially reducing the effect of high silver ion concentrations in the media [41]. Green-stage culture cells are more sensitive to stressors, and thus AgNPs reduce cell growth due to photosystem disruption, DNA, the cell’s interior and exterior damage, or light transmission reduction by the cell’s surface covered with NPs [2]. HpG culture cells are at an exponential growth phase, meaning that most cells have a relatively thin wall and are vulnerable to stressors such as AgNPs [42]. Red-stage cells have a thicker cell wall and are already adapted to stressful environmental conditions [23], and thus the impact of AgNPs on productivity was lower.

Astaxanthin synthesis results showed that concentration in HpG culture changes insignificantly related to the increase in AgNP concentration in the media. It remained constant at ~0.43% of dry weight, indicating that AgNPs are not suitable stressors as astaxanthin production enhancers. Yet, the HpR culture’s astaxanthin concentration was significantly reduced by AgNPs. It was reduced from 1.9% in the control culture to 0.6% on average in the culture containing 8 mg/L AgNPs. As well as on dry weight, various metals have different effects on astaxanthin accumulation. A high concentration of MgAC NPs (1 g/L) increased astaxanthin production by up to 40% [33]. We observed a lot of variation between different cultivation periods. Linear regression analysis shows that about half of the variation could be attributed to AgNP concentration in the media. Another half of the variation may be explained by the variation in other factors, most likely by a residual nitrogen concentration in the media, and a starter culture growth phase in the variation between cultivation periods.

Light microscopy and scanning electron microscopy images revealed cell morphology changes. As well as for dry weight, higher AgNP concentrations had a negative effect on cell counts in both HpG and HpR cultures. HpG cultures’ cells were much more susceptible to the AgNPs’ effect as all the changes in cell count and morphology were expressed more than in the HpR cultures. We observed that AgNPs might have affected cell division and reproduction. Zoospores’ cell count increased in cultures cultivated with lower AgNP concentrations (0–2 mg/L). During the *H. pluvialis* life cycle, only young cells are motile. Moreover, higher AgNP concentrations induced the formation of gametocysts. No such effect was observed in the HpR cultures, except in 8 mg/L AgNP concentration where very few gametocysts were found. The concentration of AgNPs needed to induce the formation of gametocysts in red-stage culture may be higher as cells in such culture are more resistant to negative influence from the environment. Gametocysts are sexual reproduction cells, which are rarely observed in *H. pluvialis* cultures, suggesting that high concentrations of AgNPs could be used as sexual-reproduction-initiating agents. It is known that gametocyst formation can be induced by exposure to extreme conditions (freezing, desiccation, and nutrient starvation), followed by favorable conditions [27]. This suggests that *H. pluvialis* uses sexual reproduction as a way to adapt to the rapidly changing environment, not suitable for certain cultures’ growth. *H. pluvialis* red-stage cells are less reproductive than green-stage cells. In both cultures, ruptured, dead cells, and cell wall remains were detected. It was shown that a certain silver ion concentration is sufficient to penetrate the cell wall and induce pigment leakage and cell deformations due to disturbance of cytokinesis and DNA damage [34]. High AgNP concentrations of 4–8 mg/L in the media resulted in a high count of dead cells in both cultures (HpG and HpR) We observed two kinds of dead cells. The first kind is the cell with a clearly ruptured tertiary cell wall and secondary cell wall intact. These shapes are indicative of reproductive cells (sexual and asexual) after releasing gametes or zoospores [23]. The second type is a cell without any visible cell wall ruptures, completely depigmented, and mostly a regular round cell shape. We observed no zoospores or gametes forming in these cells, indicating death caused by external factors. We observed fewer ruptured cells in the HpG culture. Green-stage cells lack a firm tertiary cell wall which could contain dead cell protoplast inside. SEM images revealed that cells cultivated in the high concentration of AgNPs were stuck together and covered by film-like substances. This can be caused by the release of polysaccharides to the surface [43] or high amounts of positive Ag^+^ ions binding to negative ligands at the cell surface, removing negative cell wall charge and allowing cells to stick [44]. It was shown that metal ions in high concentration have affected DNA and gene expression [45]. We observed cases of abnormal cell division. These abnormalities may be caused by high intracellular Ag^+^ concentration affecting cytokinesis protein function or their gene expression. Slightly abnormal shape cells were observed in all samples, including the control, but the severity and rate of deformation increased with increased AgNP concentration in the medium. Abnormally undivided cell clusters (Figure 4d) were observed only in HpG culture with concentrations 4 and 8 mg/L of AgNPs and in HpR with 8 mg/L.

Though AgNPs show a negative impact on green algae *H. pluvialis*, studies show that plants (Oriental lilies) treated with 100 mg/L AgNPs for two years have a higher leaf greenness index, form more flowers, and flower longer. It has been suggested that this positive effect of NPs might be due to enhanced antioxidant enzyme activity, which reduces oxidative stress and may reinforce plant responses to other types of stresses, such as salinity or high temperature [46].

Otherwise, our experiments show that certain AgNP amounts have a negative impact on algae growth. This opens the possibility of using AgNPs as an agent against algae growth in water containers or on surfaces coated with the AgNPs layer. Moreover, silver should not be used in algae cultivation systems as it has a negative effect on algae productivity.

## 5. Conclusions

The addition of silver nanoparticles (AgNPs) did not have any positive effect on green algae *H. pluvialis* growth or astaxanthin synthesis. Green-stage (HpG) culture’s growth was more negatively affected by higher AgNP concentrations compared to red-stage (HpR) cultures. AgNP concentration in the media did not have any significant effect on the astaxanthin production in the HpG culture and had a strong negative effect on the HpR culture. The abundance of AgNPs in the media led to significant changes in cell morphology in both the HpG and HpR cultures. A high amount of undivided and highly deformed cells was observed in both cultures. In conclusion, AgNPs could not be used to obtain higher *H. pluvialis* biomass or astaxanthin yields, but there is a possibility of inducing sexual and asexual reproduction using low amounts of AgNPs.

## Figures and Tables

**Figure 1 nanomaterials-13-01618-f001:**
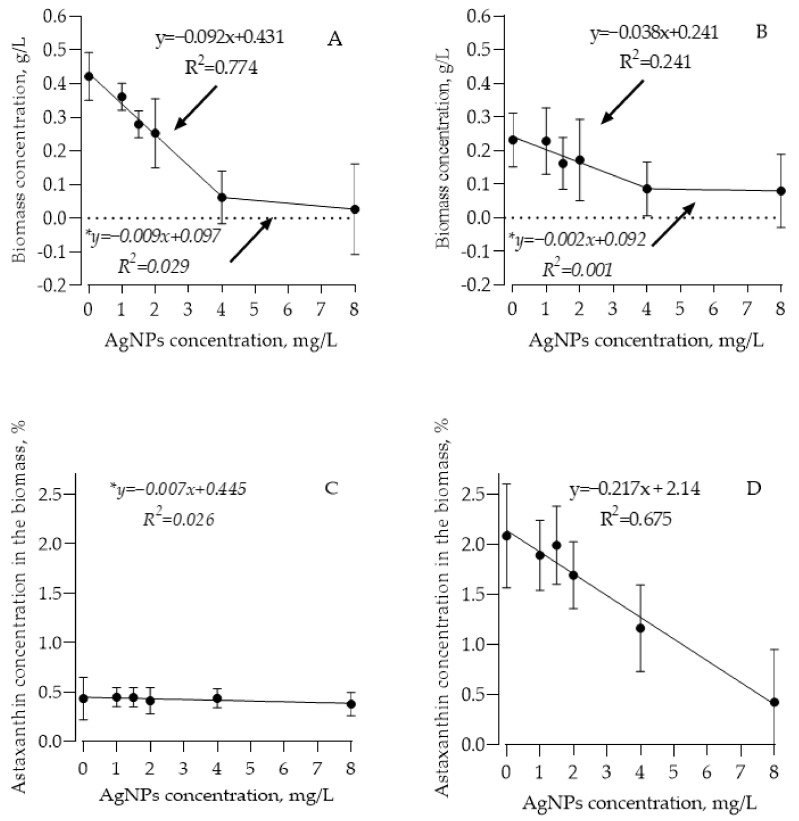
Change in biomass after seven days in the HpG (**A**) and HpR (**B**) cultures, and final astaxanthin concentration in the HpG (**C**) and HpR (**D**) cultures. Symbol “*” indicates statistically insignificant dependencies.

**Figure 2 nanomaterials-13-01618-f002:**
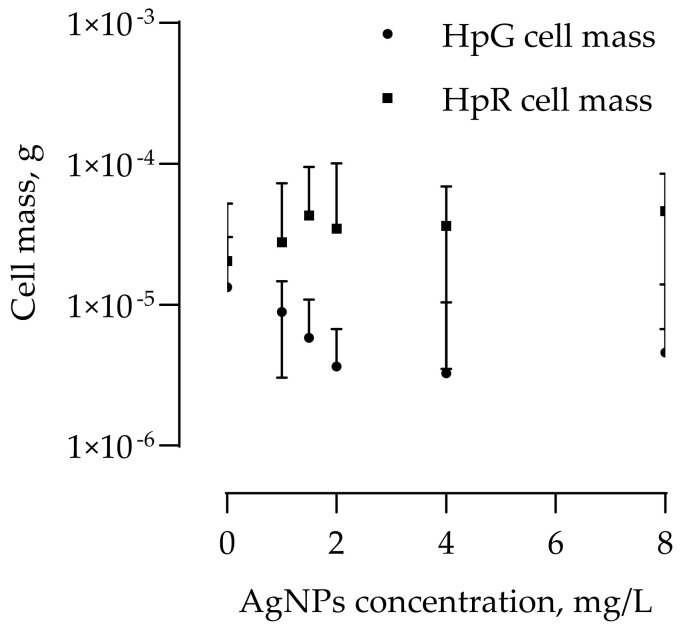
Average cell mass dependence on AgNP concentration in a medium in HpG and HpR cultures.

**Figure 3 nanomaterials-13-01618-f003:**
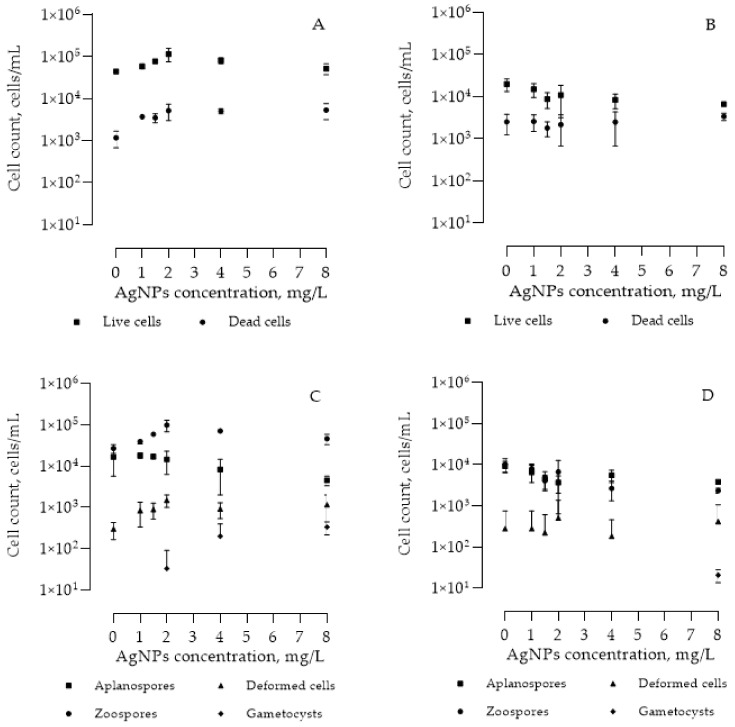
Cell count of the total amount of live and dead cells in HpG culture (**A**) and cell count of the total amount of live and dead cells in HpR culture (**B**). Cell count of different morphology cells in HpG culture (**C**) and cell count of different morphology cells in HpR culture (**D**).

**Figure 4 nanomaterials-13-01618-f004:**
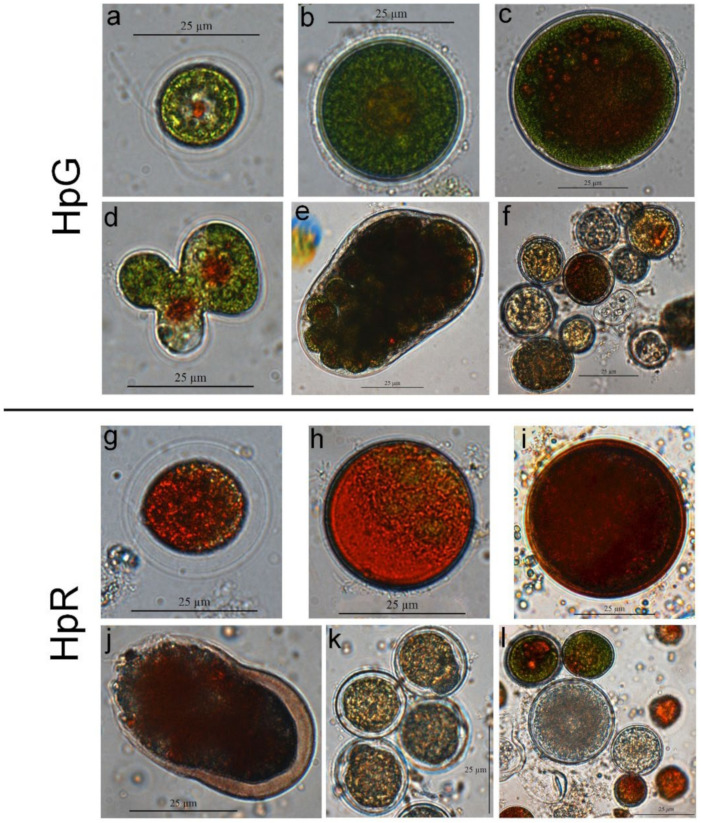
Light microscopy images of *H. pluvialis* after exposure to different concentrations AgNPs; (**a**,**g**) zoospores in the control samples; (**b**,**h**) aplanospores in the control samples; (**c**,**i**) aplanospores increased in diameter in low AgNPs concentrations (1.5–4 mg/L); (**d**) deformed and incompletely divided cell (2 mg/L); (**e**) gametocyst (8 mg/L); (**f**,**l**) cultures in high AgNP concentrations, mostly dead cells (4–8 mg/L); (**j**) ruptured cell (8 mg/L); (**k**) depigmented cells (8 mg/L).

**Figure 5 nanomaterials-13-01618-f005:**
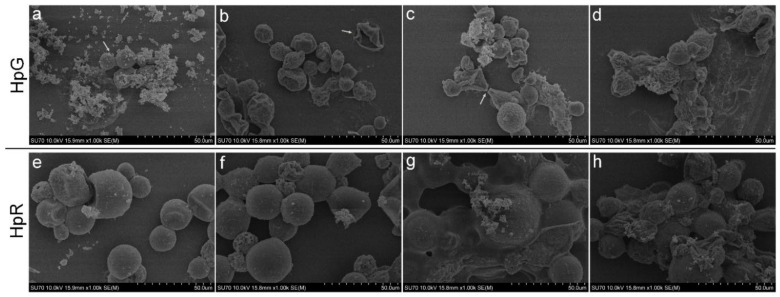
SEM analysis of *H. pluvialis* after exposure to different concentrations AgNPs. HpG (**a**–**d**) and HpR (**e**–**h**) sample images. (**a**) HpG control culture (arrow indicates cells); (**b**) HpG culture with 1 mg/L AgNPs (arrow indicates remaining cell wall after division); (**c**) HpG culture with 2 mg/L of AgNPs (arrow indicates abnormally unseparated cells); (**d**) HpG culture with 8 mg/L of AgNPs (conglomerated cells with dead cells debris); (**e**) HpR culture control; (**f**) HpR culture with 1 mg/L of AgNPs; (**g**) HpR culture with 4 mg/L of AgNPs; (**h**) HpR culture with 8 mg/L of AgNPs (conglomerated cells with dead cells debris).

**Table 1 nanomaterials-13-01618-t001:** The size distribution of AgNPs incubated for seven days in an algal culture medium.

Diameter of AgNPs, nm	%
10–20	1
20–30	7
30–40	9
40–50	14
50–60	19
60–70	11
70–80	14
80–90	5
90–100	66
>100	14

## Data Availability

The data presented in this study are available within the article.

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
