# Peer review of "Effect of Biosynthesized Silver Nanoparticles on the Growth of the Green Microalga Haematococcus pluvialis and Astaxanthin Synthesis"

_nanomaterials, 2023, doi:10.3390/nano13101618_

Round 1

Reviewer 1 Report (New Reviewer)

The paper is well and understandable written. I have only a few questions and points for the authors:

- row 106, a mistake in error (0.0.9)

- row 107, BB medium should be described (the methodic should be clear without searching in any citation)

- row 157, explain PBS (Nanomaterials is a journal not only for biologists)

- row 171, section 2.5 have to be described in detail. Without looking into citation 32 the reader does not know, how the particle distribution was distinguished. In addition, this article is not open access, for many readers is the information locked (for example me).

- row 209. Why do the authors anticipate the death of the microalgae? It could be caused by a depressed metabolism or any other effect, not leading to the organism's death. There should be described, why the authors anticipate the death. It could be helpful to explain the life cycle of the microalgae, e.g. how long does take one reproduction cycle, if the cultures were synchronized (which may play a great role in discussion), etc. Information about the life cycle, when the AgNPs were applied is mentioned only in the discussion section.

The cell death problem continues for the rest of the manuscript.

- row 307, R2 = 0.248, this does not seem like a good fit, I do not think it is possible to say "we were able to adapt a linear regression..."

- row 383, the authors should try to measure the intracellular concentration of Ag. This is not a hard task, the ICP-MS technique could be easily adapted for this reason.

The English should be slightly improved. There are sentences with unclear formulation (e.g. row 36, after citation 3; row 48), or with wrong word order (row 239 - should begin Changes in different...).

Author Response

We thank for the valuable insights and review of our manuscript.

We have provided answers to the comments in attached file.

Reviewer 2 Report (New Reviewer)

The article describes the effect of silver nanoparticles on algae. The methods used are suitable and the results are clearly presented and discussed. However, there are some minor aspects that should be precise.

(1) There is no information about minimal inhibition or algicidal concentration of nanoparticles. This is important because the authors propose a linear model including AgNPs concentration, but in figure A curve omits the last value and B doesn't. I suppose that MIC value is about 4 mg/l and the linear model could be true till the AgNPs concentration reaches the minimal inhibition/algicidal concentration and then stays in a plateau. Please add the info about MIC and calculate or describe the proper linear regression model.

(2) Please, add the equation and the value of the coefficient R2 to figure 1C.

(3) In the figure, please add the trend lines, taking into account the MIC values ​​(linear or appropriately matched to the distribution).

(4) Figure 3 needs to be rearranged so that the legends are more readable (they do not overlap the curves).

Author Response

We would like to thank reviewer for the valuable comments and revision of our manuscript. We provided the answers to reviewers comments and suggestions in the attached file.

This manuscript is a resubmission of an earlier submission. The following is a list of the peer review reports and author responses from that submission.

Round 1

Reviewer 1 Report

The manuscript by Endriukaityte and colleagues attempts to provide rare model of algae for studying silver nanoparticles (AgNPs) and their biological effects. The scientific question of the study is whether AgNPs could potentially be used for obtaining higher biomass or astaxantin from H. pluvialis algae and the conclusion is that they cannot. However, there are many concerns in the study that do not clearly support this conclusion.  

The study presents more a summary of few observations than a carefully designed and evaluated research. It is very difficult to follow the storyline due to very descriptive style of presenting the data without offering the explanatory interlineations. Despite the very interesting and important topics, especially for environmental pollution potentially caused by nanoparticles and its wide-reaching consequences, the study seems to be rather preliminary and lacks important analyses that would support its conclusions.

Essentially, the complete characteristics of AgNPs used in the study is missing. The reader is unable to learn what size/diameter were AgNPs in the culture media, whether they were colloidally stable in the media used for algae cultivation – for each concentration of AgNPS used and for the whole duration of nanoparticle exposition, which counts 7 days. This is too long time for any nanoparticles used in any study to be perfectly sure that they did not undergo aggregation (especially the high-concentrated AgNPs media), or created no stable colloidal solution and maybe were laying on the bottom of the culture glasses, or other possible modifications that need to be clearly excluded as they could underly the observed effects but do not represent specific effects of nanoparticles. This would for example explain the observed effect of 2 mg/l AgNPs concentration that caused effects (e.g., cell deformation) not observed by the higher concentrations used (4-8 mg/l). The reader also does not learn about how (at least potentially) the AgNPs elicit their effects on these algae – need to be internalized inside the cells or may just be present on the cell membrane (maybe triggering some receptors?) to stimulate the observed effects?

Authors often give statements like “the number of dead cells increase”, however any analysis showing and confirming such increase is not available in the manuscript. Also, presented light microscopy pictures and SEM pictures give no clue how big is the cell population displaying the presented effects – whole population, half population, some cells… No analysis to support the statements about changed cell shape, number of deformed cells, more debris (how much more?), sticking cells together (all cells together, groups of cells together, few cells together?), motile cells (it is concluded that lower count of motile cells appeared due to AgNPs), dead cells (concluded high count of dead cells due to AgNPs), etc. – no assays and quantifications available.

Authors conclude that AgNPs may have negative influence on cell growth but no cell growth was measured and quantified. The same effect on cell division – no cell division was measured and quantified. The main aim of the study was to measure the productivity of the cultures; however, authors admit that culture itself created a main factor influencing productivity – then how was it able to estimate the portion of production AgNPs were responsible for? Given the conclusion that AgNPs reduce the productivity of the algae cultures…

Altogether, the topic of the study is very interesting and it is definitely worth to investigate the object of the study, but in the present form the manuscript is too preliminary and more of analyses are required to confirm presented interpretations and conclusions.

Author Response

Thank you for the review of our manuscript.

Please see the attachment for responses to your comments.

Reviewer 2 Report

Authors examine the potential use of silver nanoparticle in the use as a stress agent in the cultivation of microalga Haematococcus pluvialis and the following production of astxanthin.

There is an obvious critical omission throughout the manuscript which is the lack of any information regarding any characteristic of the produced nanoparticles. I understand that this is not the focus of the study but a comprehensive approach should include also these kind of data that would validate the quality of the used nanoparticles.  Editor should evaluate this critical important omission. 

Other minor issues:

 Lines 37-39: In the introduction regarding the mode of action of the silver nanoparticles some more details would be useful aiming to show the kind of complexity in the field. Metal nanoparticles show a broad spectrum of activity, and this should be reflected in the introduction.

Line 44: avoid don’t, doesn’t etc, use precise scientific language.

Line 65: correct “has” to “have”.

Line 66: “has” to “have”.

Line 83: “the lethal effect on the organism remains the same” what does same lethal effect mean?

Please use AgNPs as abbreviation for the plural form for the nanoparticles throughout the manuscipt.

Line 210-211: as a future reference it would be useful and informative to show microscopy images with mixed live+dead population in high AgNPs levels. And maybe a quantification and association between the levels of the dead algae and the levels of AgNPs.

Line 219 delete “a”.

Discussion and conclusion should be carefully checked for the level of the English language quality and style.

Line 280-283: this is a hypothesis, and this is the way that it should be presented since no experimental data are provided in order to support this as a statement.

Author Response

Thank you for the review of our manuscript.

Please see the attachment for the response.

Round 2

Reviewer 1 Report

The revised manuscript by Endriukaityte and colleagues contains some new information, however, this information did not improve the scientific outcome of the presented experimental study. Reviewer´s notes why is it so are following:

 We have added Materials and methods section “2.6. Characterization of silver nanoparticles” and Results section “3.1. Extracellular biosynthesis and characterization of obtained silver nanoparticles” regarding the comments.

With all due respect, reviewer cannot agree that scanning electron microscopy can analyze nanoparticle size and distribution as authors report in the methods. It only can estimate the shape of nanoparticle core and potentially pre-estimate the size of few nanoparticles in the view, as it cannot “see” anything else from the metal-based nanoparticles. DLS analysis is the one to determine the diameter of nanoparticles in the preparation and biological solutions (they will be different), as well as their real size distribution and zeta-potential in these solutions – at least for the duration of the experiment – meaning here 7 days. Only like this is the nanomaterial used well characterized. These all data are essential to understand with what kind of nanomaterial the experiments are done to be able to compare the data with the world-wide scientific literature of bio-nanomaterials. Otherwise, the data are of low scientific value as no one can understand what nanomaterial has been used, what properties the nanomaterial possesses and the observed effects are linked to some unknown not characterized material and therefore not reproducible for others.

 If there were some degree of agglomeration of NPs in the culture it was not an issue as we see clear negative linear effect of AgNPs increased concentration on HpG culture biomass production (Fig. 3a) and HpR culture astaxanthin concentration (Fig. 3d). There is negative linear effect pattern in HpR biomass concentration (Fig. 3b) and HpG astaxanthin concenteation (Fig 3c), but it is not statistically significant.

I am not sure what expertise authors have to work with nanomaterials, however aggregation of nanoparticles is actually a great issue in the field of nanomaterials in biological applications. Aggregates of nanoparticles cannot be considered nanomaterials, as such belong to the micro-range (or higher range) and the effects cannot be anymore attributed to the nanomaterial-induced biological activity, but whatever other kind of a material what aggregated material represents. That authors observed some effects may of course be valid, however if these effects are triggered by pieces of aggregated material, they are not caused by nanomaterial, so the authors should reconsider their statements about nanoparticles and report the material as microparticles or according to the size of aggregates. As SEM picture clearly shows that this is the case here (reviewer was already pointing earlier that it seems to be an issue here – it was somehow clear from the results), the interpretation of the results about AgNPs affecting biomass production or astaxanthin concentration cannot be done for AgNPs as a nanomaterial, however for Ag-aggregated micro-material (or other higher-range material). Which excludes them from the nanomaterial research.

Comment:The reader also does not learn about how (at least potentially) the AgNPs elicit their effects on these algae – need to be internalized inside the cells or may just be present on the cell membrane (maybe triggering some receptors?) to stimulate the observed effects?

Response: Introduction part improved in regard to the comment

With all the respect to the authors, no other study, mentioned or not in introduction, can explain the particular experimental setting of this particular study. What is known about nanoparticles in the nanomaterial community is important, however whether it is applicable for presented experimental approaches here is not sure unless it is proved. This part of the study is missing here. Moreover, given that in this study with high probability the effects on algae are induced by aggregated Ag material it is conceivable that it has not been internalized by algae, therefore it could not follow the path presented in introduction. There are many ways to confirm the Ag metal in the tissue (plant or algae or animal) and only then one can suggest what potential mechanisms (in-cell or extra-cell) underly observed effects. Unless proved, the reader can guess what from the available possibilities listed in the published literature may be the correct here. And this is not what a reader expects from the scientific paper. Is the material in experiment at the beginning colloidally stable? Show it. Is it aggregated at some point during experiment? Show it. Is it internalized into the cells? Show it. Is it not? Show it. Are cells more viable? Show it. Are less viable? Show it. Show anything what has been discussed. Only what is shown can be understand and analyzed by the reader. Publications of others are only to learn what others found out and serve to compare “our” results to “theirs” to broaden the common understanding in the respective science field.

Unfortunatelly we are unable to provide numerical values for different kind of cells (motile cells, aplanospores, gametocysts, dead cells etc.) our main goal was to determine what kind of effects AgNPs could have on algae. We agree that it would be very useful to have such data to improve our research.

Reviewer is surprised that authors are unable to count the cells. Usually, it is the basic operation in culture studies, as number of living/dead cells is essential for estimation of whatever experimental outcome. It is not the same to apply certain number of nanoparticles in a precise volume of a medium to three or three thousand cells. This has to be precisely established to be able to repeat the same experiment reliably. Therefore, the authors´ statement that actually the main goal was to determine what kind of effects AgNPs could have sounds bizarre as knowing the number of living/dead/affected/damaged/changed…etc. cells is essential to validate any biological effect at all. Just an example – any substance for plants or any medicament comes not only with the potential effects listed, however also with probability how often these effects/side effects are expected to happen. Without this knowledge, the substance would be worthless and the whole use of such substance would be impossible. Given that authors also present what effects their experimental setting may cause, these effects have to be supported by at least some kind of a number value for scientific community to understand what is in this particular establishment to be expected.

We have provided data of dry weight changes in the culture after 7 days of cultivation in various AgNPs concentrations. Although we cannot provide cell count data (cells/ml or dividions/day), but we provided changes in the biomass productivity and astaxanthin concentration (Fig. 3). Amount of the biomass and astaxanthin is the measurements which are the most valuable from the economic perspective of algae cultivation.

Reviewer does not question that these outcomes are the most valuable from the economic perspective of algae cultivation, however reviewer gives a notice that in light of the presented data in this particular study, these outcomes are not to be attributed to Ag as nanomaterial and are not to be interpreted as effects caused by Ag nanoparticles. Whether these effects are consequence of other kind of aggregated Ag material, which also can be valid for economic perspective is not a task of this reviewer.

Round 3

Reviewer 1 Report

Author Response: Our main objective was to show that Geobacillus sp. bacteria can be adapted to the synthesis of AgNPs and their effects on H. pluvialis cells. We determined the size and shape of our AgNPs using SEM, also we were able to find authors who determined the particle size using SEM ((The scanning electron microscopy (SEM, S-3400N; HITACHI, Tokyo, Japan) was used to determine the shape and size of ZnO NPs). https://doi.org/10.7717/peerj.758

Reviewer: It makes no sense to discuss the basics of nanoparticle characterization. Authors are invited to read the book from Joachim Wegener – Measuring Biological Impacts of Nanomaterials (Springer), where they will find how the nanomaterials must be characterized for biological experiments (as well as in many other relevant sources online). There are many techniques described, with light scattering as the recommended one, but the SEM is definitely not one of them (at all not like a single method used for characterization), regardless of what few authors published. Also, authors will find how the biological experiments measuring impacts of nanomaterials should be designed, analyzed and interpreted to offer a reliable picture about nanomaterial-induced biological effects. This applies to all responses of authors to any reviewer´s previous comments. If the material entering the experiment is not well characterized, if the experiment is not analyzed appropriately, it is impossible to expect that conclusions of such study have scientific impact. Authors may see the effects, however, whether observed effects are attributable to their nanomaterial is without already mentioned proofs highly questionable. Science is based on proofs not on suggestions, even if they are right. Reviewer understands that authors believe what they observed. However, they need to persuade the reader, that what they did corresponded to norms of such experimental approaches and based on this, the reader (other scientist) can be confident to build their own research. This is impossible for any reader of any bio-nanomaterial community from the presented study in the present form. Reviewer therefore kindly encourage the authors to redo their study, add all necessary proofs that are essential for estimation and interpretation of nanomaterial-mediated biological effects and believe that then their research surely will be well-taken by the nanomaterial community.